# Towards Personalized Antioxidant Use in Female Infertility: Need for More Molecular and Clinical Studies

**DOI:** 10.3390/biomedicines9121933

**Published:** 2021-12-17

**Authors:** Jan Tesarik

**Affiliations:** MARGen Clinic, Camino de Ronda 2, 18006 Granada, Spain; jtesarik@clinicamargen.com; Tel.: +34-606-376-992

**Keywords:** aging, antioxidant, female fertility, ovary, oxidative stress, personalization, pregnancy, reductive stress, uterus

## Abstract

Treatment with antioxidants is increasingly used to slow down aging processes in different organs of the human body, including those implicated in female fertility. There is a plethora of different natural, synthetic or semi-synthetic medicines available on the market; most of them can be purchased without medical prescription. Even though the use of antioxidants, even under conditions of auto-medication, was shown to improve many functions related to female infertility related to oxidative stress, the lack of medical control and supervision can lead to an overmedication resulting in an opposite extreme, reductive stress, which can be counterproductive with regard to reproductive function and produce various adverse health effects in general. This paper reviews the current knowledge relative to the effects of different antioxidants on female reproductive function. The persisting gaps in this knowledge are also highlighted, and the need for medical supervision and personalization of antioxidant prescription is underscored.

## 1. Introduction

Oxidative processes within cells are normally localized in mitochondria, where they are coupled to mechanisms that convert chemical energy from oxygen molecules or nutrients into adenosine triphosphate (ATP). ATP is subsequently used as a source of energy for a variety of cellular processes, while part of energy not incorporated into ATP is released as waste products, called reactive species, mainly reactive oxygen species (ROS) [1]. Some ROS are used in the regulation of cell signaling and homeostasis [2], and those in excess are scavenged by antioxidant defense systems, mainly acting in mitochondria, the organelles where the ROS are generated [3]. An imbalance between oxidants and antioxidants in favor of the oxidants leads to a disruption of redox signaling and control, and causes cellular damage. This condition is referred to as “oxidative stress” [4]. Oxidative stress leads to mitochondrial dysfunction and massive release of ROS from mitochondria to cytosol, where they generate lipid peroxidation affecting the integrity of cell membranes [5] as well as protein, RNA, and DNA oxidation leading to apoptosis [6]. 

In different types of cells of the female reproductive system, various etiological factors cause pathological reactions that eventually converge to oxidative stress [7]. In case assisted reproduction techniques (ART) are employed, they may create another layer of oxidative stress-related cell damage, superimposed on that already present in the explanted oocytes and their associated somatic cells. This occurs because the cell manipulation and in vitro culture conditions created in a clinical laboratory setting cannot recreate the exact conditions under which natural fertilization takes place [8,9]. 

In order to combat oxidative stress, many types of treatment with externally administered antioxidants have been proposed over the past decades. Basically, treatment strategies using antioxidants employ two general modes of administration: first, oral supplementation and, second, in vitro supplementation via media for gamete and embryo culture (only applicable in case ART techniques are used). Some clinically used antioxidant medicines (e.g., vitamins C and E) act as direct ROS scavengers, while others act indirectly, by stimulating intracellular signaling pathways leading to increased production of endogenous antioxidant systems (e.g., growth hormone), or possess both the direct and the indirect activity (e.g., melatonin) [7]. Many treatment protocols using different antioxidants in vivo or in vitro have been described. Many studies showed an improvement of fertility outcomes with the use of antioxidants, whereas others did not find any effect. This may reflect unequal sensibilities of different parts of the female reproductive system to the treatment, age-related and inter-individual differences, as well as effects of associated comorbidities. More importantly, the indiscriminate and uncontrolled self-medication of antioxidants, based on the reasoning “the more the better”, can even be dangerous for the patients’ health in general. This danger has been treated very insufficiently in the literature.

This review summarizes the current experience with the use of antioxidants in the treatment of female infertility, as related to the female age, the part of the reproductive system affected, the type and mode of application of the antioxidant used and the presence of different comorbidities. Data presented suggest an urgent need for establishing personalized treatment strategies adapted to each particular condition.

## 2. Materials and Methods

An electronic search of the PubMed, Embase, Medline, and MedRxiv databases from 1990 to 2021 was conducted. A combination of medical subject headings and keywords was used to generate a subset of citations as following (in alphabetical order): (i) aging; (ii) antioxidant; (iii) female fertility; (iv) ovary; (v) oxidative stress; (vi) personalization; (vii) pregnancy; (viii) reductive stress; (ix) uterus. 

The primary search was performed following Preferred Reporting Items for Systematic reviews and Meta-Analyses (PRISMA) guidelines, Refs. [10,11] using the exclusion criteria recommended. However, owing to the paucity of data published on this topic, and in order to encourage rapid publication of new relevant findings, no publication was discarded, and those only marginally related to the topic also were included.

## 3. Results

This section presents data concerning the effects of different antioxidant treatments on the function of individual organs of the female reproductive system under different pathological conditions. Its individual subsections are focused on the ovarian factor, including ovarian aging and corpus luteum function; the uterine factor, including the placental function and the uteroplacental transition; and pregnancy complications. The conditioning effects of comorbidities on antioxidant efficiency and the possible ways of reaching data necessary for personalized antioxidant use conclude this section. 

### 3.1. Ovarian Factor

#### 3.1.1. Ovarian Factor in Young Women

Polycystic ovary syndrome (PCOS) and endometriosis are the leading ovarian causes of infertility in young women. Both PCOS and endometriosis are associated with oxidative stress [12]. 

Oxidative stress in PCOC patients is related to an increased serum prolidase activity, which also appears to be associated with increased cardiovascular risk and menstrual irregularities [13]. PCOS patients also show increased oxidative stress markers in follicular fluid and decreased oocyte and embryo quality [14], as well as a high level of chronic inflammation markers [15]. Little data are available as to the utility of externally administered antioxidants to reduce oxidative stress and improve fertility in PCOC patients (Table 1) [16]. The administration of an antioxidant cocktail containing vitamin A, vitamin B1, vitamin B6, vitamin B12, vitamin C, vitamin D3, vitamin E, nicotinamide, and folic acid significantly improved pregnancy rates in PCOS patients [17]. Two randomized controlled trials (RCTs) were performed with in vivo antioxidant treatment of PCOS patients, and they showed an improvement of oocyte and embryo quality after treatment with resveratrol [17], and an increase in implantation, pregnancy, and cumulative pregnancy rates after treatment with vitamins D and E [18], respectively (Table 1). Moreover, encouraging results were obtained in an animal model (rat) by using mitochondria-targeted antioxidant therapy [19]. Interestingly, administration of growth hormone (GH) during ovarian stimulation of young PCOS patients led to a significantly improved pregnancy, clinical pregnancy and spare embryo cryopreservation rate while reducing the total number of oocytes recovered and thus the patients’ discomfort (Table 1) [20]. However, factors other than antioxidant actions could also have contributed to these GH effects.

As to endometriosis, RCTs showed that in vivo treatment of patients with vitamins C and E reduced pelvic pain and decreased the concentration of peritoneal fluid inflammatory markers as compared with untreated patients (Table 1). The study failed to detect any positive effect of in vivo treatment of patients with endometriosis with vitamin C on in vitro fertilization (IVF) outcomes [21], while resveratrol supplementation during IVF cycles even appeared to be detrimental to pregnancy outcomes (Table 1) [22]. Further studies are needed to find out whether other combinations of antioxidant medicines could also improve IVF outcomes, in addition to reduce pain and inflammation, in patients with endometriosis.

#### 3.1.2. Ovarian Factor in Older Women

Even though the causes of ovarian factor infertility present in young women usually persist until more advanced ages their relative importance decreases, while the main causes of infertility in older women are related to ovarian aging [7]. Diminished ovarian reserve (DOR), characterized by the decline in the quality and quantity of oocytes, is the most significant feature of ovarian aging and becomes the main reason for infertility and ART failure [7]. DOR is at the origin of primary ovarian insufficiency (POI) and can be accentuated by a variety of age-unrelated factors [7], including chromosome X structural abnormalities and X-autosome translocations [23], single-gene perturbations ( some located on the X-chromosome and others on autosomes [24,25]), mendelian disorders implicated in other pathologies [24], mutations of genes (both nuclear and mitochondrial ones) affecting mitochondrial function [26,27], and disturbances of cell signaling pathways, especially those involved in cell protection against oxidative stress [28,29]. Independently of the exact cause, DOR and POI converge to mitochondrial damage, oxidative stress, and diminished ATP production, leading to inflammation, apoptosis, and telomere shortening [30], in addition to a decrease in the production of estradiol by the granulosa cells [31], a condition known to reduce oocyte developmental potential by causing an imbalance between nongenomic effects of androgens and estrogens at the oocyte surface [32]. 

In view of the above considerations, the use of antioxidants appears justified to improve oocyte quantity and quality in older women. Different antioxidants were tested in animal models of ovarian aging [33], and most of them gave encouraging results (Table 2). In mice, oral administration of vitamins C and E prevented the aging-related negative effects on ovarian reserve, metaphase II oocyte chromosomal aberrations and oocyte apoptosis in mice [32], N-acetyl-L-cystein mitigated age-related reduction of litter size and increased telomerase activity and telomere length [33], coenzyme Q10 restored oocyte mitochondrial function and fertility of aged animals [34], and melatonin improved age-induced fertility decline by attenuating ovarian mitochondrial oxidative stress [35]. Based on these encouraging data, different antioxidants were introduced into the treatment protocols used for ART in older women (Table 2). However, data obtained in humans are scarce and inconsistent, mainly because of the superimposition of different effects of the agents used, the heterogeneity of the patient populations studied, and the paucity of randomized controlled trials with appropriate controls. 

An RCT performed in older women showed that the treatment of women aged >40 years with growth hormone resulted in an improvement of delivery and live birth rates [36]. Numerous subsequently published studies confirmed these findings and extended the use of growth hormone to other female infertility indications [37], including some younger women with previous unexplained IVF failures [38,39,40,41]. Growth hormone was also shown to improve human oocyte in vitro maturation from the germinal vesicle to metaphase II stage [42]. On the other hand, growth hormone can also fail to improve IVF outcomes in some women, and several clues were suggested to distinguish those patients who would benefit from growth hormone treatment from those who would not [43]. Moreover, even though growth hormone is known to stimulate cell signaling pathways involved in the defense against oxidative damage [7], it is not known whether, and to what extent, this effect is responsible for the improvement of the oocyte developmental potential described in the above studies. 

Melatonin was also used to improve reproductive function of aging women. In a randomized and placebo-controlled clinical study, the treatment of women 43–49 years old with melatonin for 6 months led to a significant decrement of serum FSH and LH levels, suggesting a recovery of pituitary function towards a more juvenile pattern of regulation [44]. The concentration of melatonin in the follicular fluid was shown to be positively correlated with antral follicle count, oocyte quality, and IVF outcomes [45,46]. A recent study, based on network pharmacology, demonstrated a multi-target mechanism of action of melatonin against DOR [47]. 

Antioxidants were also suggested to improve the corpus luteum function in animal models [48]. However, experience with the use of antioxidants to rescue the corpus luteum from premature luteolysis is still insufficient. 

### 3.2. Uterine Factor

In the mouse model [49], the age-related decrease in uterine receptivity for embryo implantation was related to a redox imbalance that could be mitigated by intake of two different antioxidants, the nicotinamide adenine dinucleotide phosphate oxidase (NOX) inhibitor apocynin and the superoxide dismutase mimetic 4-hydroxy-2,2,6,6,-tetramethylpiperidinyloxy (TEMPOL). The former also significantly increased litter size and restored decidua thickness, while the latter increased birth weight, and both decreased protein carbonylation level in the uterus of reproductively aged mice [49]. Even though age-related decrease in the uterine and placental function were also described in some women, ART outcomes of aging women obtained with the use of oocytes from young donors are excellent [50], suggesting that uterine aging only represents a marginal problem in humans. Moreover, in a large cohort study, and after adjusting for age, no significant association between DOR and a short luteal phase was detected [51]. 

On the other hand some studies described age-unrelated problems of uterine receptivity leading to recurrent implantation failure (RIF). RIF usually refers to failure to achieve a clinical pregnancy after transfer of at least four good-quality embryos in a minimum of 3 fresh or frozen cycles in woman under the age of 40 years [51,52]. There is little information about the possible role of oxidative stress in the etiology of RIF. It was shown that co-treatment of women with the history of RIF with growth hormone during preparation for the transfer of embryos obtained with oocytes from young donors fertilized with normal sperm significantly improved embryo implantation, pregnancy, and live birth rates via beneficial actions on endometrial receptivity [53]. However, the mechanism of this growth hormone effect, including the question of whether and how this effect was mediated by the hormone antioxidant action, was not addressed. 

### 3.3. Unexplained ART Failure Not Related to Age

Most of the studies related to the use of different antioxidants in the treatment of female infertility were carried out in cases of previous unexplained ART failures [54]. After exclusion of cases with poor semen quality, this group of patients can be subdivided into two subgroups: those showing poor oocyte and embryo quality and those with apparently normal oocytes and embryos. 

In both conditions, the use of different antioxidants improved the outcomes in most, though not all cases (Table 3). A prospective cohort study [55] failed to demonstrate a significant improvement of ART outcomes after oral treatment of women with vitamin C during ovarian stimulation. Similarly, in an RCT using oral treatment with vitamin C for 14 days beginning with the day of oocyte retrieval, no improvement of implantation and clinical pregnancy rate was achieved [56]. As to vitamin E, another direct antioxidant agent, an observational study demonstrated a positive correlation of endogenous vitamin E concentrations in serum and follicular fluid with the number of metaphase II oocytes and some morphological parameters of embryo quality after IVF [57], but experience with therapeutic use of exogenous vitamin E is currently lacking. 

More conclusive results were reported with the use of Coenzyme Q10 (CoQ10), a dual-role (pro-oxidant and antioxidant) molecule whose reduced form (ubiquinol) protects biological membranes from lipid peroxidation by recycling vitamin E and acting as an antioxidant [58]. In two RCTs, in vivo treatment of women with CoQ10 before the beginning of ovarian stimulation was shown to result in lower aneuploidy rates in embryos generated by IVF [59] and higher numbers of large antral follicles and higher fertilization rates [60], respectively. However, the use of CoQ10 did not increase cumulative pregnancy rate and live birth rate as compared with controls [60]. On the other hand, an RCT evaluating the effects of the addition of CoQ10 to oocyte in vitro maturation (IVM) medium did not find any significant differences in the oocyte maturation and postmeiotic aneuploidy rates in oocytes from women <30 years old as compared with controls, while IVM in the presence of CoQ10 improved oocyte maturation and decreased blastocyst aneuploidy in women aged 38–46 years [61]. 

The use of resveratrol, a natural polyphenol of plant origin with antioxidant and anti-inflammatory properties, in women gave conflicting results. On the one hand, when added to IVM medium in an RCT, it increased the maturation rate and improved meiotic spindle morphology and chromosome arrangement in human oocytes cultured from the germinal vesicle to the methaphase II stage [62]. On the other hand, a retrospective study evaluating the in vivo effect of resveratrol added during the IVF cycle showed a diminished cumulative pregnancy rate and an increased miscarriage rate as compared to untreated women [63]. 

Two prospective cohort studies [64,65] and four RCTs [66,67,68,69] evaluated in vivo effects of melatonin administered from the cycle preceding the IVF attempt. It was shown that melatonin reduced the number of degenerated oocytes [64,69], increased the number of mature oocytes [66,69], and increased fertilization rate [65] and embryo quality [69]. One RCT [68] failed to detect any significant differences in cumulative pregnancy rate or oocyte and embryo parameters, but the authors recognized that the study was not sufficiently powered. Another RCT failed to demonstrate an effect of melatonin added to IVM medium on in vitro maturation of human germinal vesicle oocytes, although melatonin appeared to improve oocyte cytoplasmic maturation and subsequent IVF clinical outcome [70]. 

**Table 3 biomedicines-09-01933-t003:** Antioxidants in the treatment of age-unrelated ART failure.

Antioxidants	Type of Study	Outcome	References
Vitamin C	Prospective cohort	No effect on ART outcomes	[55]
	RCT	No improvement of IR and CPR in ART	[56]
Vitamin E	Observational, non-interventional	Positive correlation of serum and FF levels with oocyte maturity and embryo quality	[57]
Coenzyme Q10	RCT	Less aneuploidy in IVF embryos	[58]
	RCT	Increased FR No effect on CPR and LBR	[59]
Resveratrol	Retrospective	Decreased CPR	[62]
		Increased MR	[63]
Melatonin	Prospective cohort	Fewer degenerated oocytes	[64]
	Prospective cohort RCT	Increased FR Improved oocyte and embryo quality	[65] [68]

Abbreviations: ART: assisted reproduction technology; IR: implantation rate; CPR: cumulative pregnancy rate; IVF: in vitro fertilization; FF: follicular fluid; MR: miscarriage rate.

### 3.4. Pregnancy

Oxidative stress is suspected to be involved in the pathogenesis of hypertensive and metabolic disorders of pregnancy [69]. A mild level of oxidative stress is a natural response of the maternal organism to pregnancy, owing to the increased metabolic activity in the placental mitochondria, related to a higher metabolic demand of the growing fetus [70]. However, above certain limits the pregnancy-associated oxidative stress becomes harmful to pregnancy and creates conditions that can cause detrimental effects to both the mother and the fetus [71]. The pathological conditions caused by excessive oxidative stress during pregnancy include hypertensive disorders, including hypertension in pregnancy, chronic hypertension, white-coat hypertension, masked hypertension, gestational hypertension preeclampsia, proteinuria, and hemolysis with elevated liver enzymes and low platelet count [72]. Moreover, oxidative stress is also involved in the pathogenesis of gestational diabetes mellitus [73,74]. 

Experience with the use of exogenous antioxidants in the prevention and treatment of these pregnancy complications is limited. Of particular interest is the possibility of using melatonin in the treatment of preeclampsia [75,76], since reduced melatonin levels were shown to be associated with the development of this pathological condition [77]. Actually, administration of melatonin to women with severe early onset preeclampsia significantly mitigated maternal endothelial oxidative injury and prolonged pregnancy [78].

Other antioxidants used for the management of preeclampsia include vitamins C and E, and resveratrol. Surprisingly, two large RCTs testing vitamin C and E [79,80] in this indication did not show any significant improvement. Better outcomes were reported for resveratrol, which was found to be an efficient supplement of oral nifedipine treatment to control blood pressure in women with preeclampsia [81,82]. 

## 4. Discussion

The treatment of pathologies associated with oxidative stress has a long tradition, but there still remain quite a lot of unanswered questions about how to draw the best benefit from the treatments used without causing undesired side effects. The point is that, unlike many other pathological conditions, oxidative stress-related diseases cannot be successfully treated by a simple elimination of the causing agent—the ROS. In fact, the ROS are natural actors in multiple cellular metabolic processes that are disturbed when they are depleted. Thus, rather than a radical action against the ROS, the therapy has to be designed so as to achieve an equilibrium in ROS production, use, and elimination of their excess load (Figure 1) [82]. 

As shown in Figure 1, the redox status of each cell depends on an interplay between ROS generation through oxidation of different substrates; their use in various physiological processes that may be more or less demanding, depending on the type and physiological status of the cell in question; and the elimination of excess ROS by means of various scavenging mechanisms based on the action of antioxidants. When the supply of antioxidants is insufficient, the excess unscavenged fraction of ROS is causing oxidative stress (Figure 1A). An equilibrium between the antioxidant levels and the amount of excess ROS leads to an equilibrated redox balance, required for the optimal cell function (Figure 1B). If, on the contrary, the supply of antioxidants exceeds the needs for ROS scavenging, reductive stress results (Figure 1C). The adaptation of antioxidant treatments is to be aimed at the achievement of the equilibrated redox balance, avoiding both extreme conditions —oxidative and reductive stress (Figure 1). 

To achieve this goal, many different aspects of each individual patient have to be taken into account. These aspects include the age, personal and familial medical history, and the nature of the disease for which the use of antioxidant treatment is considered as well as any significant comorbidities. Only a synthetic view of all these aspects can result in an appropriate choice of the type and dose of the antioxidant(s) to be used.

### 4.1. Choice of the Antioxidant(s) Used

There are many types of antioxidants available on the market, and the choice of the most appropriate one(s) is not an easy task. Some antioxidants, such as vitamins C and E, are direct ROS scavengers, while others, such as growth hormone, act indirectly by stimulating intracellular pathways leading to the production of endogenous antioxidants. Melatonin has a particular position among antioxidants because it combines the properties of both a direct and an indirect antioxidant agent [7]. 

Since some antioxidants also exert other effects, some of which may be beneficial for potentially existing comorbidities, this reasoning can prevail in the decision-taking process. The age of the patient and the history of her previous ART attempts can also be used as criteria. For instance, growth hormone was shown to increase the delivery and live birth rates in women aged > 40 years [36], to improve IVF outcomes in young women in whom poor oocyte and embryo morphology had been associated with a previous IVF failure [39], and to affect positively IVF outcomes in women with PCOS [18]. As for melatonin, it was suggested as a possible noninvasive treatment of adenomyosis and endometriosis [83], based on its inhibitory effect on 17β-estradiol-induced migration, invasion and epithelial–mesenchymal transition in normal and endometriotic human endometrial epithelial cells [84]. Moreover, it was proposed, owing to its pan-antiviral activities, to be used as an adjuvant accompanying COVID-19 and other antiviral vaccines [85,86]. Before more sophisticated criteria, based on precision medicine, are available, the pragmatic approach taking into account the age and comorbidities of the treated patients may thus be of help in the antioxidant choice.

It is evident that the choice of the most suitable antioxidant for therapeutic purposes will necessarily depend on the chemical formula of that specific compound, its structure giving it its unique properties, and its mode of action [87]. 

### 4.2. Choice of the Appropriate Antioxidant Dose

In the 1990s, when the use of antioxidants to prevent and treat diseases related to oxidative stress became popular, the prevailing idea was “the more the better”. Nonetheless, the current experience with the use of antioxidants has challenged this concept [82]. Actually, rather than considering the treatment of oxidative stress on the “all-or-none” basis, recent data converge to the concept of “redox imbalance”, which can be caused by oxidative stress as well as by an opposite extreme, referred to as “reductive stress” (Figure 1). The counterproductive effect of excess antioxidant medication, leading to reductive stress, has been clearly demonstrated in men in whom antioxidant treatment was indicated to improve sperm DNA integrity [88]. 

Apart from the direct adverse effects of excess antioxidant doses on the pathological conditions treated, the resulting reductive stress is as harmful as oxidative stress for human health in general and is implicated in many pathological processes at the origin of various diseases unrelated to the pathology for which antioxidant treatment has been indicated [89]. As a matter of example, redox imbalance resulting in reductive stress was implicated in mitochondrial dysfunction leading to cardiomyopathy [90] and in an impairment of the immune system function increasing the likelihood of developing malignancies by means of decreased immune protection and direct cell and DNA damage [91]. Paradoxically, chronic consumption of antioxidant supplements, such as vitamins or flavonoids, may result in a pro-oxidant effect [92] that can contribute to a variety of diseases, including cardiomyopathy [93], pulmonary hypertension [94], muscular dystrophy [95], Alzheimer’s disease [96]. Parkinson’s disease [97], insulin resistance and metabolic syndrome [98], rheumatoid arthritis [99], and cancer [100]. 

Experience with different types of antioxidants suggests that the risk of an overdose leading to reductive stress may vary among them [101,102]. For instance, most cases of reductive stress-related issues were reported for β-carotene, and vitamins C and E [92], whereas melatonin lacks any serious side-effects even at doses highly exceeding those currently used to combat oxidative stress [103]. Further research is needed to explain these kinds of differences. All of these issues are expected to be resolved, hopefully in the near future, by the application of the “precision medicine” approach, whereby the treatment of each patient will be based on the evaluation of the precise nature of her redox regulation and the elucidation of the importance and necessity of the precise nature of redox regulation from three aspects: differences in redox status, differences in redox function, and differences in the effects of redox therapy [104]. According to the opinions published by experts of two important reproductive societies, the European Society of Human Reproduction and Embryology [105], and the American Society for Reproductive Medicine [106], the subject of the use of antioxidants in the prevention and treatment of female infertility is important but is still surrounded by many unresolved questions that need to be urgently dealt with.

## 5. Conclusions and Future Directions/Alternatives

The administration of antioxidants can be an effective tool with which to improve female infertility associated with oxidative stress. However, it is important to take into account the individual condition of each patient in order to determine the type of antioxidant used, alone or in combination, the duration of treatment and the doses administered. Inadequate choices can lead to an effect that is opposite to that desired. In addition, excess antioxidant treatment, in general, can cause reductive stress that can lead to the development of various pathological conditions concerning the patient’s health in general. Medical supervision, based on the individual condition of each patient, is advised in order to choose the optimal antioxidant(s) to be used at adequately adapted doses.

## Figures and Tables

**Figure 1 biomedicines-09-01933-f001:**
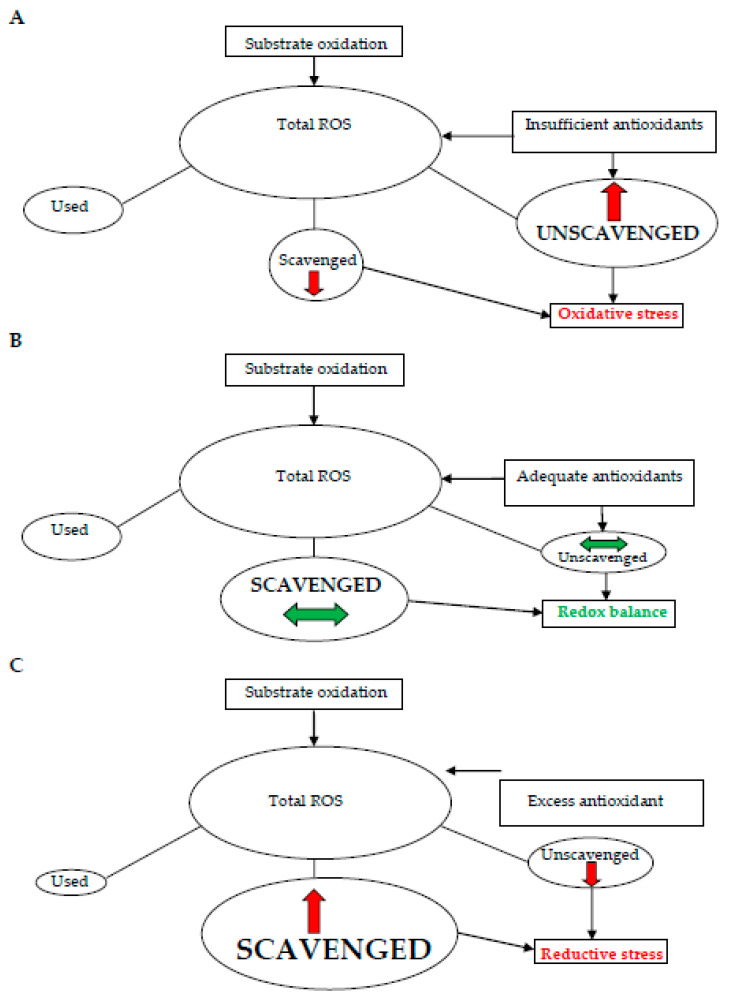
Schematic representation of the cellular redox status under the conditions of oxidative stress (**A**), adequate redox balance (**B**), and reductive stress (**C**).

**Table 1 biomedicines-09-01933-t001:** Antioxidants in the treatment of ovarian factor in young women.

Antioxidants	Pathology	Outcome	References
Resveratrol	PCOS	Improvement of oocyte and embryo quality	[16]
Vitamins D and E	PCOS	Increase in IR, PR, and CPR	[18]
Mixture of vitamin A, vitamin B2, vitamin B6, vitamin B12, vitamin C, vitamin D3, vitamin E, nicotinamide, and folic acid	PCOS	Increase in PR	[17]
Growth hormone	PCOS	Increase in PR, CPR, and embryo cryopreservation rate	[20]
Vitamins C and E	Endometriosis	No effect on IVF outcome	[21]
Resveratrol	Endometriosis	Detrimental for IVF outcome	[22]

Abbreviations: PCOS: polycystic ovary syndrome; IR: implantation rate; PR: pregnancy rate; CPR: cumulative pregnancy rate; IVF: in vitro fertilization.

**Table 2 biomedicines-09-01933-t002:** Antioxidants in the treatment of age-related ovarian factor.

Antioxidants	Species	Outcome	References
Vitamins C and E	Mouse	Protection of ovarian reserve Prevention of oocyte chromosomal aberrations and apoptosis	[33]
N-acetyl-L-cystein	Mouse	Increased litter size, telomerase activity, and telomere length	[34]
Coenzyme Q10	Mouse	Restoration of oocyte mitochondrial function Improvement of fertility	[35]
Melatonin	Mouse	Improvement of ovarian mitochondrial function and fertility	[36]
	Human	Recovery of pituitary function Improved oocyte quality and IVF outcomes	[37,38,39]
Growth hormone	Human	Improvement of DR and LBR	[38,39]
Growth hormone	Human	No effect in some women	[40]

Abbreviations: DR: delivery rate; LBR: live birth rate; IVF: in vitro fertilization.

## Data Availability

No unpublished data is included.

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
