# Peer review of "Towards Personalized Antioxidant Use in Female Infertility: Need for More Molecular and Clinical Studies"

_biomedicines, 2021, doi:10.3390/biomedicines9121933_

Round 1

Reviewer 1 Report

The author investigated how personalized antioxidant use is effective in female infertility treatment. Please see my suggestions below.

L56-60. Please make this aim of the study more relevant. What makes special this study? Which is its novelty character or its special aspects? Why have the author chosen this topic? What differentiate this paper from others in the same topic?

Instead of paragraph L62-66 (which is very poor related how the literature was selected), a PRISMA flow chart is recommended to be done; please see both Page at al. papers, where this type of graphic is very well described: Page, M.J.; McKenzie, J.E.; Bossuyt, P.M.; Boutron, I.; Hoffmann, T.C.; Mulrow, C.D.; Shamseer, L.; Tetzlaff, J.M.; Akl, E.A.; Brennan, S.E.; et al. The PRISMA 2020 statement: An updated guideline for reporting systematic reviews. Journal of Clinical Epidemiology 2021, 134, 178-189, doi:10.1016/j.jclinepi.2021.03.001. Page, M.J.; McKenzie, J.E.; Bossuyt, P.M.; Boutron, I.; Hoffmann, T.C.; Mulrow, C.D.; Shamseer, L.; Tetzlaff, J.M.; Moher, D. Updating guidance for reporting systematic reviews: development of the PRISMA 2020 statement. Journal of Clinical Epidemiology 2021, 134, 103-112, doi:10.1016/j.jclinepi.2021.02.003. The paper will have a much more professional aspect. Please also mention ALL criteria used for the papers' selection. You will see all of them in the PRISMA flow chart provided as model in the papers above recommended. Do not forget mentioning about medical subject headings terms (MeSH) whici are usually used for searching in PubMed and embase subject heading (Emtree) for searching in Embase.

Tables 1 and 2. I am sure that the author can find more relevant literature to sustain/support these Tables and to add more information as well. (Yeung A.W.K., et al. Antioxidants: scientific literature landscape analysis. Oxid. Med. Cell. Longev., 2019 https://doi.org/10.1155/2019/8278454 ; Tit D.M., et al. Somatic-vegetative symptoms evolution in postmenopausal women treated with phytoestrogens and hormone replacement therapy. Iran. J. Public Health 2017, 46(11), 1128-1134)

In the Discussion section, adding a summarising figure, which to describe the most relevant characteristics of antioxidants in the topic will boost the relevance of the paper. 

Moreover, subsection 4.1. must be completed that: The choice of the most suitable antioxidant for therapeutic purposes necessarily depends on the chemical formula of that specific compound, its structure giving it its unique properties, implicitly its mode of action. - as it is stated by Glevitzky I., et al. Statistical Analysis of the Relationship Between Antioxidant Activity and the Structure of Flavonoid Compounds. Rev. Chim. 2019, 70(9), 3103-3107. https://doi.org/10.37358/RC.19.9.7497

Please improve last section as 5. Conclusions and future directions/alternatives, revealing what must be done further to increase antioxidants relevance in the topic. The titles of subsections can be renamed in the Review type papers as it is more suitable for the content.

Author Response

Comment:

L56-60. Please make this aim of the study more relevant. What makes special this study? Which is its novelty character or its special aspects? Why have the author chosen this topic? What differentiate this paper from others in the same topic?

Response:

In order to the aim of this study more relevant, I have added a warning against potential dangers of indiscriminate and uncontrolled self-medication with antioxidant at the end of the last but one paragraph of the introduction, also mentioning that this danger has been treated very insuficintly in the literature

Comment:

Instead of paragraph L62-66 (which is very poor related how the literature was selected), a PRISMA flow chart is recommended to be done; please see both Page at al. papers, where this type of graphic is very well described: Page, M.J.; McKenzie, J.E.; Bossuyt, P.M.; Boutron, I.; Hoffmann, T.C.; Mulrow, C.D.; Shamseer, L.; Tetzlaff, J.M.; Akl, E.A.; Brennan, S.E.; et al. The PRISMA 2020 statement: An updated guideline for reporting systematic reviews. Journal of Clinical Epidemiology 2021, 134, 178-189, doi:10.1016/j.jclinepi.2021.03.001. Page, M.J.; McKenzie, J.E.; Bossuyt, P.M.; Boutron, I.; Hoffmann, T.C.; Mulrow, C.D.; Shamseer, L.; Tetzlaff, J.M.; Moher, D. Updating guidance for reporting systematic reviews: development of the PRISMA 2020 statement. Journal of Clinical Epidemiology 2021, 134, 103-112, doi:10.1016/j.jclinepi.2021.02.003. The paper will have a much more professional aspect. Please also mention ALL criteria used for the papers' selection. You will see all of them in the PRISMA flow chart provided as model in the papers above recommended. Do not forget mentioning about medical subject headings terms (MeSH) whici are usually used for searching in PubMed and embase subject heading (Emtree) for searching in Embase.

Response:

I have included the recommended PRISMA method (including the suggested references) in the revised manuscript.

Comment:

Tables 1 and 2. I am sure that the author can find more relevant literature to sustain/support these Tables and to add more information as well. (Yeung A.W.K., et al. Antioxidants: scientific literature landscape analysis. Oxid. Med. Cell. Longev., 2019 https://doi.org/10.1155/2019/8278454 ; Tit D.M., et al. Somatic-vegetative symptoms evolution in postmenopausal women treated with phytoestrogens and hormone replacement therapy. Iran. J. Public Health 2017, 46(11), 1128-1134)

Response:

I have added the suggested references to the revised manuscript.

Comment:

In the Discussion section, adding a summarising figure, which to describe the most relevant characteristics of antioxidants in the topic will boost the relevance of the paper. 

Response:

A figure summarising the combinations of factors leading, to oxidative stress, balanced redox state, and reductive stress, respectively, has been added to the Discussion section.

Comment:

Moreover, subsection 4.1. must be completed that: The choice of the most suitable antioxidant for therapeutic purposes necessarily depends on the chemical formula of that specific compound, its structure giving it its unique properties, implicitly its mode of action. - as it is stated by Glevitzky I., et al. Statistical Analysis of the Relationship Between Antioxidant Activity and the Structure of Flavonoid Compounds. Rev. Chim. 2019, 70(9), 3103-3107. https://doi.org/10.37358/RC.19.9.7497

Response:

The suggested text has been added to the revised manuscript.

Comment:

Please improve last section as 5. Conclusions and future directions/alternatives, revealing what must be done further to increase antioxidants relevance in the topic. The titles of subsections can be renamed in the Review type papers as it is more suitable for the content.

Response:

The section 5 has been renamed as suggested

Reviewer 2 Report

The manuscript review existing data about antioxidant treatments to improve female fertility and live birth rate. My conclusion from this review is that while such treatments are easy to provide, the medications are commonly taken by patients for various reasons, and do not need medical prescriptions, available studies are not easily comparable, not well designed, and consequently the conclusions are not easy to make.

In summary, the message of the paper is that there are no robust conclusions from available studies, and better studies are needed, which is in contradiction with the title of the manuscript, “Towards personalized antioxidant use in female infertility”. 

I also did not see in the Discussion the opinion of reproductive societies, which usually provide the opinion of experts, in such treatments. 

Author Response

Comment:

The manuscript review existing data about antioxidant treatments to improve female fertility and live birth rate. My conclusion from this review is that while such treatments are easy to provide, the medications are commonly taken by patients for various reasons, and do not need medical prescriptions, available studies are not easily comparable, not well designed, and consequently the conclusions are not easy to make.

In summary, the message of the paper is that there are no robust conclusions from available studies, and better studies are needed, which is in contradiction with the title of the manuscript, “Towards personalized antioxidant use in female infertility”. 

Response:

The title has been changed to “Towards personalized antioxidant use in female infertility: need for more molecular and clinical studies”

Comment:

I also did not see in the Discussion the opinion of reproductive societies, which usually provide the opinion of experts, in such treatments. 

Response:

I have added the published opinions of two important reproductive societies, ESHRE and ASRM, together with the two corresponding references. Both societies agree as to the importance of the topic, but they also recognize the need for more studies to go ahead.

Round 2

Reviewer 1 Report

The authors responded to my requests.

Reviewer 2 Report

revised appropriately.